# Antileishmanial and Antiplasmodial Activities of Secondary Metabolites from the Root of *Antrocaryon klaineanum* Pierre (Anacardiaceae)

**DOI:** 10.3390/molecules28062730

**Published:** 2023-03-17

**Authors:** Gabrielle Ange Amang à Ngnoung, Lazare S. Sidjui, Peron B. Leutcha, Yves O. Nganso Ditchou, Lauve R. Y. Tchokouaha, Gaëtan Herbette, Beatrice Baghdikian, Theodora K. Kowa, Desire Soh, Raoul Kemzeu, Madan Poka, Patrick H. Demana, Xavier Siwe Noundou, Alembert T. Tchinda, Fabrice Fekam Boyom, Alain M. Lannang, Barthélemy Nyassé

**Affiliations:** 1Department of Chemistry, Faculty of Science, University of Maroua, Maroua P.O. Box 814, Cameroon; 2Laboratory of Phytochemistry, Centre for Research on Medicinal Plants and Traditional Medicine, Institute of Medical Research and Medicinal Plants Studies, Yaoundé P.O. Box 13033, Cameroon; 3Bioorganic and Medicinal Chemistry Laboratory, Department of Organic Chemistry, Faculty of Science, University of Yaoundé I, Yaoundé P.O. Box 812, Cameroon; 4Natural Product and Environmental Chemistry Group (NAPEC), Department of Chemistry, Higher Teachers’ Training College, University of Maroua, Maroua P.O. Box 55, Cameroon; 5Laboratory of Medicinal Chemistry & Pharmacognosy, Department of Organic Chemistry, Faculty of Science, University of Yaoundé I, Yaoundé P.O. Box 812, Cameroon; 6Laboratory of Pharmacology and Drugs Discovery, IMPM, Yaoundé P.O. Box 13033, Cameroon; 7Aix-Marseille Univ, CNRS, Centrale Marseille, FSCM, Spectropole, Campus de St Jérôme-Service 511, 13397 Marseille, France; 8Aix Marseille Univ, CNRS 7263, IRD 237, Avignon Université, IMBE, 27 Blvd Jean Moulin, Service of Pharmacognosy, Faculty of Pharmacy, 13385 Marseille, France; 9Department of Chemistry, Higher Teacher Training College Bambili, The University of Bamenda, Bambili, Bamenda P.O. Box 39, Cameroon; 10Antimicrobial and Biocontrol Agents Unit, Laboratory for Phytobiochemistry and Medicinal Plants Studies, Department of Biochemistry, Faculty of Science, University of Yaounde 1, Yaounde P.O. Box 812, Cameroon; 11Department of Pharmaceutical Sciences, School of Pharmacy, Sefako Makgatho Health Sciences University, Pretoria 0204, South Africa; 12Department of Chemical Engineering, School of Chemical Engineering and Mineral Industries, University of Ngaoundere, Ngaoundere P.O. Box 454, Cameroon

**Keywords:** anacardiaceae, *Antrocaryon klaineanum*, cerebroside, antileishmanial activity, antiplasmodial activity

## Abstract

*Antrocaryon klaineanum* is traditionally used for the treatment of back pain, malaria, female sterility, chlamydiae infections, liver diseases, wounds, and hemorrhoid. This work aimed at investigating the bioactive compounds with antileishmanial and antiplasmodial activities from *A. klaineanum*. An unreported glucocerebroside antroklaicerebroside (**1**) together with five known compounds (**2**–**6**) were isolated from the root barks of *Antrocaryon klaineanum* using chromatographic techniques. The NMR, MS, and IR spectroscopic data in association with previous literature were used for the characterization of all the isolated compounds. Compounds **1**–**4** are reported for the first time from *A. klaineanum.* The methanol crude extract (**AK-MeOH**), the *n-*hexane fraction (**AK-Hex**), the dichloromethane fraction (**AK-DCM**), the ethyl acetate fraction (**AK-EtOAc**), and compounds **1**–**6** were all evaluated for their antiparasitic effects against *Plasmodium falciparum* strains susceptible to chloroquine (3D7), resistant to chloroquine (Dd2), and promastigotes of *Leishmania donovani* (MHOM/SD/62/1S). The **AK-Hex**, **AK-EtOAc**, **AK-MeOH**, and compound **2** were strongly active against Dd2 strain with IC_50_ ranging from 2.78 ± 0.06 to 9.30 ± 0.29 µg/mL. Particularly, **AK-MeOH** was the most active—more than the reference drugs used—with an IC_50_ of 2.78 ± 0.06 µg/mL. The **AK-EtOAc** as well as all the tested compounds showed strong antileishmanial activities with IC_50_ ranging from 4.80 ± 0.13 to 9.14 ± 0.96 µg/mL.

## 1. Introduction

In the past decade, interest in traditional medicinal plants has increased exponentially with the arrival of new pathogens, resistance to existing drugs, climate change, and amidst all of this, the side effects of some synthetic drugs [1,2]. In Cameroon, malaria is present throughout the country and leishmaniasis in the northern regions; they are the two main diseases caused by tropical insects with the highest number of deaths in the last 10 years. Moreover, malaria represents the most widespread disease, with about 6.7 million cases and 13 839 deaths in 2021 [3]. Primarily, the number of people dying from malaria and leishmaniasis is due to the length of treatment, low efficacy, high cost, numerous side effects, induction of parasite resistance and, above all, vaccines that will not be available soon [1,4]. Thus, the research and promotion of local traditional medicinal plants as a source of new and natural medicines has become a high priority [2,4,5,6]. Anacardiaceae is a family of plants known for their nutritional and therapeutic importance. Anacardiaceae plants are composed of trees or shrubs with about 81 genera grouped into 800 species including the genus *Antrocaryon* and the Antrocaryon klaineanum Pierre species [7]. In Cameroon, three main species of the genus *Antrocaryon* are well known: *Antrocaryon guillaumin, A. micraster*, and *A. klaineanum*. Antrocaryon klaineanum is called “White mahogany (English)”, “Bouton d’antilope (French)” [8]. It is a tree about *35* m high that is well known by the population for the primary healthcare treatment for ailments such as back pain, malaria, female sterility, chlamydiae infections, liver diseases, wounds, etc. [7,8]. This work focused on the search for bioactive compounds and was motivated by previous studies conducted on A. klaineanum which mentioned the presence of a panel of bioactive secondary metabolites well known for their antiplasmodial activities. Previous phytochemical study of *A. klaineanum* have shown that flavonoids, ellargic acid, gallic acid, steroids, triterpenoids, and saponins are the main classes of secondary metabolite found in the genus Antrocaryon as well as in *Antrocaryon klaineanum* [7,8,9]. To the best of our knowledge, only the antiplasmodial activities of *Antrocaryon klaineanum* have already been reported [7,9]. This research work focuses on the chemical study and the antiplasmodial and antileishmania activities of the root barks of *A. klaineanum*. The chemical study led to the isolation and the characterization of six secondary metabolites, among which an unreported glucocerebroside derivative (**1**), together with five known compounds (**2**–**6**)—among which two coumarins (**2**–**3**), an ellagic acid (**4**), and two steroids (**5**–**6**). The compounds **1**–**4** are reported for the first time from *A. klaineanum.* The methanol crude extract (**AK-MeOH**), the *n-*hexane fraction (**AK-Hex**), the dichloromethane fraction (**AK-DCM**), the ethyl acetate fraction (**AK-EtOAc**) and compounds **1**–**6** were all evaluated for their antiparasitic effects against *Plasmodium falciparum* strains susceptible to chloroquine (3D7) and resistant to chloroquine (Dd2) and promastigotes of *Leishmania donovani* (MHOM/SD/62/1S).

## 2. Results

The methanol crude extract of the air-dried root barks of *A. klaineanum* was subjected to liquid–liquid extraction. The *n*-hexane, dichloromethane, and ethyl acetate fractions were all fractionated and purified using column chromatography (CC) over silica gels as well as using sephadex LH-20, which led to the isolation of six secondary metabolites—among which an unreported glucocerebroside, antroklaicerebroside (**1**), together with five known compounds (**2**–**6**) (Figure 1). The structures of all these compounds were determined based on the interpretation of NMR and MS analysis in comparison to reported data.

### 2.1. Structure Elucidation of Compound **1**

Compound **1** was isolated as a white powder with a dextrorotatory activity (${\left[\alpha \right]}_{586}^{20}$= + 0.2 (c 2.5, Acetone)), m.p. 184–186 °C. Molecular formula C_48_H_93_NO_10_ was deduced from its HR-ESI-MS spectrum which shown a sodium adduct peak [M+Na]^+^ at *m/z* 866.6694 (calcd. *m/z* 866.6692 for C_48_H_93_NO_10_Na^+^) accounting for three double bond equivalents (Appendix A). The UV spectrum revealed maximum absorptions at 269 nm (Appendix A). The IR spectrum (Appendix A) of compound **1** revealed some characteristic absorption bands of hydroxyl (3 324 cm^−1^), amide (3 269 cm^−1^), *trans*-double bond (1 618 and 963 cm^−1^), and aliphatic chain (2 918 and 2 850 cm^−1^) [10,11,12]. The presence of these functional groups was in agreement with atoms found in the molecular formula and further supported by the 1D- and 2D-NMR.

The ^1^H-NMR and HSQC spectra of compound **1** (Table 1, S7, S8, S11, and S12) displayed the signals attributable to olefinic protons [*δ*_H_ 5.41 (1H, m) and 5.35 (1H, m)], a *β*-D-glucopyranosyl moiety [*δ*_H_ 4.34, 1H, d, *J* = 7.8 Hz, anomeric proton; (*δ*_C_ 104.7 (1″), 75.0 (2″), 78.0 (3″), 71.6 (4″), 77.9 (5″), and 62.7 (6″)], an amide linkage [*δ*_H_ 7.51, 1H, d, *J* = 8.9 Hz (NH); *δ*_C_ 177.2 (1′)], a nitrogen-bearing methine [*δ*_H_ 4.27, 1H, m; *δ*_C_ 51.6 (2), three oxygenated methines at *δ*_H_ 3.60 (1H, t, *J* = 6.0 Hz)/*δ*_C_ 75.6 (3), *δ*_H_ 3.53 (1H, m)/*δ*_C_ 72.9 (4), and *δ*_H_ 4.02 (1H, dd, *J* = 7.7–3.8 Hz)/*δ*_C_ 73.0 (2′), and an oxygenated methylene *δ*_H_ 4.05 (1H, dd, *J* = 10.7–6.2 Hz)/*δ*_H_ 3.82 (1H, dd, *J* = 10.7–3.9 Hz)/*δ*_C_ 69.9 (1). The above information is in agreement with an unsaturated dihydrosphingosine-type glucocerebroside derivative [11,12,13,14,15]. It was found that there were two aliphatic chains according to the integration of the protons assigned to the terminal methyls at *δ*_H_ 0.88 (t, *J* = 7.0 Hz, 6H). The positions of all the hydroxyl groups found on the dihydrosphingosine chains were supported by the COSY (Figure 2, Appendix A) spectrum, which reveals cross-correlation peaks between protons at *δ*_H_ 3.56 (H-3) with protons at *δ*_H_ 4.30 (H-1) and 3.58 (H-4), which was also in agreement with the HMBC spectrum. Furthermore, the connectivity between the sugar moiety with the aglycone as well as the connection between the two main chains were supported by the HMBC (Figure 2, Appendix A), which revealed a correlation between the anomeric proton (*δ*_H_ 4.30) with the only oxymethylene carbon (*δ*_C_ 70.2) of the aglycone and between the amidomethine proton (*δ*_H_ 4.34) of the sphingosine unit with the carboxyl (*δ*_C_ 177.2) of the fatty acid fragment. These features further indicated a glucosphingosine-type cerebroside derivative of compound **1** [10,13,14,15].

The lengths of the sphingosine and the fatty acid chains as well as the position of the double bond were identified based on the information given by the MS/MS spectrum (Appendix A) in addition to LR-ESI-MS (−ve/+ve). Then, the sodium adduct [M+Na]^+^ *m/z* 866.7 cation was collided in the MS/MS experiments and the fragmentation spectrum was recorded for a collision energy of 65 eV. By using the precursor ion of elemental composition of C_48_H_93_NO_10_Na^+^ as an internal standard, exact mass measurements could be made for the main fragment ions of interest. Among the different peaks observed on the MS/MS spectrum, the fragment of the fatty acid was detected at [M+Na]^+^ *m/z* 500.3 for C_24_H_47_NO_8_Na as proposed in the formation mechanism (Appendix A). From the above formation, the length of the sphingosine moiety in which the double bond is found was deduced. In addition, the position of the double bond was firstly proposed at Δ^18^ by the MS/MS spectrum showing an ammonium adduct with the loss of an equivalent of *n*-non-1-enyl [M-C_9_H_17_+NH_4_]^+^ at *m/z* 467.3 (calcd. *m/z* 467.3) supported by the fragment at *m/z* 510.3 [M-H_2_O+Sugar+C_20_H_38_NO_2_+Na]^+^ for C_26_H_49_NO_2_Na (calcd. *m/z* 510.3) (Appendix A). Other useful fragments were also found on the MS/MS spectrum at *m/z* 528.4 [C_26_H_49_NO_7_Na^+^], *m/z* 686.6 [C_42_H_81_NO_5_Na^+^], and *m/z* 704.6 [C_42_H_81_NO_4_Na^+^]. Furthermore, the formation of the double bond at Δ^18^ is also in agreement with the consecutive formation of ion from the loss of an H_2_O molecule, characteristic of a C18 hydrocarbon stable chain with a double bond (Figure 3) [12,14,16]. On the other hand, the sphingosine chains of ceramides as well as cerobrosides are usually related to palmitoyl-CoA in co-occurrence with serine leading to a C18 aliphatic moiety [15].

The double bond detected on the sphingosine moiety is found to be in the (*E*)-configuration. This assumption is deduced from the chemical shift of the allylic carbons; when their *δ*_C_ is ranging from 27–28 ppm, the double bond is (*Z*)-configuration, while when *δ*_C_ is ranging from 31–34 ppm, the double bond is (*E*)-configuration [11,12,14,17,18,19]. The stereochemistry 2*S*, 3*S*, 4*R,* and 2′*R* of compound **1** was proposed based on the comparison of the ^13^C-NMR chemical shifts of C-2, C-3, C-4, and C-2′ with those of aralia cerebroside, plakosides C and D, and gynuramide II [13,14,20,21], which is in accordance with the natural occurring phytoceramide as well as cerebroside [22,23]. So, compound **1** was fully characterized for the first time based on its spectroscopic and spectrometric data and systematically named (2*S*,3*S*,4*R*,2′R,18*E*)N-((-3,4-dihydroxy-*β*-D-gluco-pyranosyl)oxy)hexacos-18-en-2-yl)-2-hydroxyhexadecanamide, and trivially named antroklaicerebroside (**1**).

Other compounds were characterized based on the analysis of their NMR and MS spectral data in comparison with the reported ones, as follows: 6,7-dihydroxycoumarin or esculetin (**2**) [24], 7-hydroxy-6-methoxycoumarin or scopoletin (**3**) [25,26], 3,3′-dimethylellagic acid (**4**) [27], and a mixture of *β*-sitosterol (**5**) and stigmasterol (**6**) [28].

### 2.2. Physico-Chemical Characteristics of Compounds **1**

Antroklaicerebroside (**1**) white powder, mp 184–186 °C, ${\left[\alpha \right]}_{586}^{20}$ = +0.2 (c 2.5, Acetone), UV(MeOH) λmax (log ξ): 269, 399, and 651 nm, IR (KBr, cm^−1^): 3324, 3269, 2918, 2850, 1618, and 963, ^1^H-NMR (Acetone-*d*_6_ and CD_3_OD, 600 MHz), and ^13^C-NMR (Acetone-*d*_6_ and CD_3_OD, 150 MHz), see Table 1; HR-ESI-MS at *m/z* 866.6694 [M+Na]^+^ (calcd. *m/z* 866.6692 for C_48_H_93_NO_10_Na^+^).

### 2.3. Antiplasmodial, Antileishmanial, and Cytotoxicity Activities of Extract, Fractions, and Compounds from Antrocaryon Klaineanum

For this purpose, the crude extract, fractions, and isolated compounds from *Antrocaryon klaineanum* were assessed in vitro for their antileishmanial and antiplasmodial activities against *L. donovani* promastigotes 1S (MHOM/SD/62/1S) and both *P. falciparum* chloroquine-sensitive (*Pf*3D7) and chloroquine-resistant (*Pf*Dd2) strains (Table 2, Figure 4 and Figure 5). They were also assessed for cytotoxicity on Raw 264.7 macrophage cells (Table 2). Based on the antiplasmodial activity classification criteria of Rasoanaivo et al. [29], the crude extract (**AK-MeOH**) strongly (IC_50_ < 5 μg/mL) inhibited the growth of the chloroquino-resistant *P. falciparum* strain (*Pf*Dd2) with an IC_50_ value of 2.78 ± 0.06 μg/mL, but was not active on the chloroquino-sensitive *P. falciparum* strain (*Pf*3D7, IC_50_ >100 μg/mL) (Table 2, Figure 4). **AK-MeOH** also exerted a selective activity against *Pf*Dd2 (SI > 179.86) with a non-deleterious effect on macrophages RAW 264.7 (IC_50_ value > 500 μg/mL). Previous investigation of the methanolic stem and root bark extracts from *A. klaineanum* collected in the central region of Cameroon (Mount Eloumden) reported a very weak activity against *P. falciparum* 3D7 (IC_50_ > 25 μg/mL) [30]. This discrepancy in activities as compared with the current result might be related to the plant’s origin, which is also correlated to the secondary metabolites content [31]. In another study, dichloromethane extract from the stem bark of *A. klaineanum* displayed an antiplasmodial potential against the chloroquine-sensitive (3D7), with an IC_50_ value of 16.7 μg/mL [9], thereby corroborating our findings to a certain extent. Except from the dichloromethane fraction (**AK-DCM**), all other fractions of **AK-Hex** and **AK-EtOAc** respectively displayed a strong antiplasmodial activity against the chloroquino-resistant *P. falciparum* strain (*Pf*Dd2) (IC_50_ < 5 μg/mL) and a high selectivity toward Raw cells (SI > 94.34 and SI > 100.40).

From these fractions (AK-Hex, AK-DCM, and AK-EtOAc), an unprecedented cerebroside (**1**) and known compounds (**2**–**6**) were isolated. Few studies report the phytochemical analysis of this plant species. However, plants belonging to the Anacardiaceae family were previously reported to contain phytosterols such as *β*-sitosterol, stigmasterol, and phenolic compounds like scopoletin (*Pseudospondias macrocarpa*) [32]. Regarding their activity, the new antroklaicerebroside (**1**) exhibited a moderate and selective antiplasmodial activity against *Pf*3D7 with an IC_50_ value of 42.04 ± 0.08 µg/mL and a selectivity index greater than 11. The phenolic compound 3,3′-dimethylellagic acid (**4**) and the coumarin-related compound scopoletin or 7-hydroxy-6- methoxycoumarin (**3**) were found to be non-active against both *Pf*3D7 and *Pf*Dd2 (IC_50_ > 50 µg/mL). Meanwhile, esculetin or 6,7-dihydroxycoumarin (**2**) displayed a moderate activity against *Pf*3D7 and a promising activity against *Pf*Dd2 with IC_50_ values of 45.57 and 9.3 µg/mL, respectively. Amongst the tested phytosterol compounds, only *β*-sitosterol exhibited a moderate antiplasmodial activity against the chloroquino-sensitive *Pf*3D7 strain. Globally, with selectivity indexes greater than 10, all the test samples were found to be selective toward Raw 264.7 cells. Conversely, the methanol (AK-MeOH) root extract of *Antrocaryon klaineanum* showed no antileishmanial activity (IC_50_ > 100 μg/mL). Further investigation of the later has resulted in more potent antileishmanial fractions, with increase in activity against *L. donovani* promastigotes. The ethyl acetate fraction (**AK-EtOAc**) exhibited the highest potency (IC_50_ = 4.80 ± 0.13 μg/mL), followed by the hexane fraction (**AK-Hex**; 15.78 ± 0.82 μg/mL and a relatively high selectivity (SI > 104.17, SI > 31.69, respectively) (Table 2). Compound (**1**) isolated from this plant selectively has showed moderate activity against *L. donovani*, with IC_50_ of 6.19 ± 0.79 µg/mL. Except esculetin (**2**), which was not active (IC_50_ > 50 μg/mL) on *L. donovani*, the other known compounds displayed good antileishmanial activity with IC_50_ values ranging from 5.56 ± 0.74 to 9.14 ± 0.96 μg/mL, with *β*-sitosterol being the most active (IC_50_ 5.56 ± 0.74 μg/mL). The identified compounds mostly belong to metabolites reputed for their biological activities and with known antiparasitic modes of action. For instance, Mogana et al. [33] reported an approximately same range of activity for scopoletin isolated from the leaves of *Canarium patentinervium* (Burseraceae) with an IC_50_ of 163.3 µg/mL against *Leishmania donovani* strain MHOM/IN/1983/AG83, an activity that was associated to its anticholinesterase activity (IC_50_ 51 µg/mL). In another study, this coumarin-related compound displayed the highest antileishmanial activity with an IC_50_ value of 6.804 µg/mL [34]. The phenolic compound esculetin (**2**) is reputed for its antioxidant, anticoagulant, antibacterial, anti-inflammatory, and antidiabetic effects, according to Liang et al. [35]. On the other hand, *β*-sitosterol from *Ifloga spicata* (asteraceae) exhibited significant activity against leishmania promastigotes (IC_50_ values of 9.2 ± 0.06 μg/mL) by targeting the DNA, inducing apoptosis, and through strong inhibition of two important leishmanial enzymes—namely, leishmanolysin and trypanothione reductase [36]. Later on, *Corchorus capsularis* L. leaf-derived *β*-sitosterol was found to exert its antileishmanial effects against *Leishmania donovani* by targeting trypanothione reductase and disrupting the parasite redox balance via intracellular ROS production in promastigotes [37]. 3,3′-dimethylellagic acid is a bioactive polyphenolic compound naturally present as a secondary metabolite in many plants. Indeed, a bioassay guided fractionation of *Hymenostegia afzelii* and *Endodesmia calophylloides* identified 3,3′-dimethylellagic as the highest antiplasmodial compound with an IC_50s_ of 4.27 μM and 1.36 μM against *Pf* 3D7 and *Pf*Dd2, respectively [38]. Moreover, ellagic acid exhibited an in vitro effect against *Leishmania donovani.* Interestingly, the same compounds significatively reduced parasitemia during an in vivo antileishmanial mouse model [39,40].

Finally, taken together, the current results are consistent with those reported in the literature, thereby confirming the traditional use of this plant.

## 3. Discussion

Using repeated chromatographic columns, six (1–6) compounds were isolated from the methanol crude extract of the root barks of Antrocaryon klaineanum Pierre (Anacardiaceae), including an unreported cerebroside (1) derivative, along with five known compounds: two coumarins (2–3), an ellargic acid (4), and two steroids (5–6). The presence of coumarins, gallic acid, as well as steroids in the Antrocaryon genus or in the Antrocaryon klaineanum species has been previously reported [30,41]. In addition, coumarins and gallic acid have been reported to be a reliable chemophenotype of the Antrocaryon genus [7]. However, this is the first report of cerebroside (1) in the Antrocaryon genus as well as in the Antrocaryon klaineanum species. This phytochemical study is in agreement with the chemophenetic survey on Antrocaryon klaineanum. This study also improves the knowledge on the classes of secondary metabolite which can be found in this plant species.

The antiplasmodial activities of the methanol crude extract (**AK-MeOH**), *n-*hexane fraction (**AK-Hex**), ethyl acetate fraction (**AK-EtOAc**), dichloromethane fraction (**AK-DCM**), and compounds **1**–**6** were tested on the chloroquine-sensitive 3D7 and chloroquine-resistant Dd2 strains of *Plasmodium falciparum*. The **AK-MeOH** (IC_50_, 2.78 ± 0.06 µg/mL), **AK-EtOAc** (IC_50_, 4.98 ± 0.81µg/mL), **AK-Hex** (IC_50_, 5.30 ± 0.19 µg/mL), and compound **2** (IC_50_, 9.30 ± 0.29 µg/mL) were strongly active against chloroquine-resistant Dd2 strain. The strong activity of the crude extract as well as the main fractions may be due to the synergic effect of compounds found in those extracts, and coumarins were the most active class of compounds. The above results clearly show that fractionation reduces the efficiency of the **AK-MeOH**. Moreover, **AK-EtOAc**, is the second-most active fraction, which can be due to the presence of coumarins and some hydrosoluble phenolic acids (ellargic acid) which are both classes of compounds that have shown an interesting antiplasmodial activity against Dd2 strain, as reported by Douanla et al. [9].

On the other hand, the **AK-MeOH**, **AK-EtOAc**, **AK-Hex**, and **AK-DCM** as well as compounds **1**–**6** were also evaluated for their antileishmanial activities against *Leishmania donovani*. The extract **AK-MeOH**, the fraction **AK-DCM**, and compound **2** reveal a low antileishmanial activity with an IC_50_ > 50 µg/mL while the fraction **AK-EtOAc** as well as compounds **1**,**3**–**6** have shown a strong antileishmanial activity with an IC_50_ ranging from 4.80 ± 0.13 to 9.14 ± 0.96 µg/mL (Table 2, Figure 4), with the fraction **AK-EtOAc** being the most active (IC_50_ of 4.80 ± 0.13 µg/mL). Contrarily to the antiplasmodial activity, the fractionation increases the antileishmanial activity, meaning that compounds are antagonists in **AK-MeOH, AK-Hex**, and **AK-DCM**, while the **AK-EtOAc** activity could be due to the presence of active metabolites presenting a synergic effect. It is also noted that the compounds **1** (IC_50_, 6.19 ± 0.79 µg/mL), **3** (IC_50_, 6.40 ± 0.80 µg/mL), and **6** (IC_50_, 8.07 ± 0.90 and 9.14 ± 0.96 µg/mL, respectively) strongly inhibited the progmastigotes of *Leishmania donovani* responsible for viceral leishmaniasis. In addition, the most active compound [**5** (IC_50_, 5.56 ± 0.74 µg/mL)] belongs to the class of coumarins, which is expected because coumarins act by damaging the mitochondrial membrane, causing ultrastructural changes in the leishmania parasite [42,43]. Indeed, the activity of compound **3** is stronger than that of its isomer (compound **2**) due to the presence of the methoxy group present in compound **3** which can reduce the damage of the mitochondrial membrane.

## 4. Materials and Methods

### 4.1. General Experimental Procedures

The melting point was measured on an Electrothermal Engineering Ltd. serial N° 4386. The IR spectrum was recorded in KBr, on an Alpha spectrometer (Bruker, Billerica, MA, USA) spectrophotometer. Compounds were dissolved in 85 µL of their corresponding solvent in the NMR tube (Eurisotop ref. D009B) before being analyzed in the spectrometer type Bruker Avance II+ at 600 MHz equipped with a Cryoprobe TCI operating under TopSpin 3.2.5, data processing under TopSpin 4.1.3. NMR sequences: ^1^H-zgcppr, COSY, HSQC, HMBC, ^13^C-zgpg30. LR- and HR-ESI-MS analyzes were performed with a SYNAPT G2 HDMS mass spectrometer (Waters, Milford, MA, USA) equipped with a pneumatically assisted atmospheric pressure ionization (API) source. The sample was ionized in positive electrospray mode, electrospray voltage: 2.8 kV; orifice voltage: 50 V; nebulization gas flow rate (nitrogen): 100 L/h. The sample was also ionized in negative electrospray mode under the following conditions: electrospray voltage: −2.27 kV; tension orifice: −50 V; nebulization gas flow rate (nitrogen): 100 L/h. Mass spectra (MS) were obtained with a time-of-flight (TOF) analyzer. Exact mass measurements were performed in triplicate with an external calibration. In MS/MS experiments, nitrogen was used as collision gas. Chromatographic columns were used over silica gel (0.063–0.200 mm) for the fractionation and the purification of compounds while thin layer chromatography (TLC) was used to check the purity of compounds which was spotted then visualized under UV (CN-6 UV, 254 nm and 365 nm) before spraying with ceric sulfate and heated at about 90 °C.

### 4.2. Plant Material

The root barks of *Antrocaryon klaineanum* Pierre (Anacardiaceae) were harvested by M. Olama Evouna Joseph Abraham in the Ossoebemva village of the Center Region of Cameroon on 10 July 2020. This plant had been identified in comparison with the specimen sample available at the Cameroon National Herbarium under the voucher 2124 SRF/CAM.

### 4.3. Extraction and Isolation

The air-dried root barks of *A. klaineanum* were used as plant material for this research work. Powdered plant material (4455.0 g) was extracted with methanol for 72 h in the percolator. The extract was evaporated to dryness using a rotary evaporator under reduced pressure leading to 628.4 g of crude extract. The methanol crude extract (600.0 g) was subjected to liquid–liquid extraction over hexane (10.4 g), dichloromethane (30.3 g), ethyl acetate (126.3 g), and residue. The EtOAc extract (120.0 g) was purified using repeated open column chromatography with silica gel as stationary phase with the system of *n-*hexane/dichloromethane and dichloromethane/methanol gradient, which led to the isolation of four compounds: **4** (9.3 mg), **3** (89.3 mg), **2** (19.2 mg), and **1** (3.1 mg). The *n-*hexane fraction and the dichloromethane fraction were both mixed together since their TLC profile was not so different. The purification of the combination led to the isolation of compounds **5** (201.3 mg) and **6** (901.4 mg).

### 4.4. Biological Activities

#### 4.4.1. Parasite and Cell Culture

The cryopreserved chloroquine-sensitive (*Pf*3D7-(MRA-102)), the multi-resistant (*Pf*Dd2) of *Plasmodium falciparum*, and the promastigote form of *Leishmania donovani* strains were obtained from BEI Resources (https://www.beiresources.org/ (accessed on 16 February 2019)) and are routinely cultured at the Antimicrobial and Biocontrol Agents Unit, University of Yaoundé I.

*Plasmodium falciparum* strains were maintained in 60 mm sterile petri dishes containing complete RPMI 1640 medium [RPMI 1640 (Sigma Aldrich) supplemented with 0.2% sodium bicarbonate (Sigma Aldrich), 0.5% Albumax II (Gibco), 1% hypoxanthine 100X (Gibco, Billings, MT, USA), 25 mM HEPES, and 0.04% gentamicin (Sigma Aldrich)] with fresh O^+^ erythrocytes of healthy human volunteers suspended at 4% haematocrit (*v/v*) and incubated at 37 °C in a 5% CO_2_ humidified atmosphere. The medium was replaced daily with fresh complete medium to propagate the culture. Ten percent (10%) Giemsa-stained thin blood smears were examined microscopically under immersion oil to monitor cell-cycle transition and parasitaemia.

*Leishmania donovani promastiogtes* were cultured in Medium 199 (Sigma, Darmstadt, Germany) supplemented with 10% Heat-Inactivated fetal Bovine Serum (HIFBS) (Sigma, Darmstadt, Germany) and 100 IU/mL penicillin/streptomycin. The culture was maintained in 75 cm^2^ cell culture flask at 28 °C and sub-cultured every 72 h.

RAW 264.7 cells were obtained from RIKEN BioResource Centre Cell Bank (Tsukuba, Japan) and cultured in Dulbecco’s modified Eagle medium (DMEM) containing 1% penicillin–streptomycin and 10% FBS under 5% CO_2_ and humidified atmosphere at 37 °C and sub-cultured every 72 h.

#### 4.4.2. Sample Preparation for Biological Assays

A stock solution was prepared at 100 mg/mL for each extract and fraction and at 50 mg/mL for isolated compound in 100% DMSO (Sigma–Aldrich, Munich, Germany). Master plates were then prepared by mixing 2 μL of each stock solution with 198 μL of fresh incomplete RPMI 1 640 culture medium to yield a concentration of 1 mg/mL for extract and fraction and of 0.5 mg/mL (1% DMSO) followed by a 5-fold serial dilution. Extracts and fractions were tested at concentrations ranging from 0.16 to 100 μg/mL, and isolated compounds at concentrations ranging from 0.08 to 50 μg/mL.

#### 4.4.3. Antiplasmodial Activity of Extract, Fractions, and Compounds

##### Synchronization of Parasite Culture

The use of mixed-stage cultures can enable the test molecules to interact with all the three stages (ring, trophozoite, and schizont) found during a 48-h life cycle of *P. falciparum* in culture. Moreover, starting the experiment with a synchronized ring stage culture provides the advantage of monitoring the growth inhibitory effect of test samples without a rise-up in parasitemia during the ring–trophozoite–schizont transitions. For this particular reason, parasites were synchronized at the ring stage by serial treatment with 5 % sorbitol for two days prior to each experiment with respect to Lambros and Vanderberg, 1979 [44] protocol.

##### SYBR Green I-Based Fluorescence Assay

Drug sensitivity assay was carried out in 96-well microtitration plates using the SYBR green I based fluorescence assay (Smilkstein et al., 2004) [45]. This assay is specifically based on the development of a strong fluorescence as a result of SYBR green entry and fixation to the exposed DNA in non-treated wells where cells have proliferated. The fact that human red blood host cells are anucleate allows the use of SYBR green for the proper monitoring of the growth of the malarial parasite.

The 96 wells flat-bottom assay plates with sorbitol-synchronized ring stage parasites (haematocrit: 1%, parasitaemia: 2%, 90 μL) were incubated in the presence of 10 µL of pre-diluted extracts, fractions, isolated compounds, and reference drugs [Chloroquine and Artemisinin (Sigma–Aldrich, Munich, Germany)] at different triplicate concentrations (100 µg/mL–0.16 µg/mL for extract/fractions and 50 µg/mL–0.08 µg/mL for compounds). After an incubation period of 72 h at 37 °C, 100 μL of the SYBR Green I solution prepared in a lysis buffer (0.2 µL of 10,000× SYBR Green I per mL of lysis buffer) consisting of Tris (20 mM; pH 7.5), EDTA (5 mM), saponin (0.008%; *m*/*v*), and Triton X-100 (0.08%; *v/v*) was added to each well, mixed twice gently with a multi-channel pipette, and incubated in the dark at 37 °C for 1 h. Fluorescence was measured using the microplate reader Infinite M200 (Tecan) at an excitation and emission wavelength of 485 and 538 nm, respectively. These data were normalized to percent control activity using Microsoft Excel software. The data analysis was performed with GraphPad Prism 8.0 software, fitting by nonlinear regression, and concentration–response curves were generated to determine the median inhibitory concentration (IC_50_) for each sample.

The resistance index (RI) was determined as the ratio of the IC_50_ of the samples on the multi-resistant strain Dd2 and the IC_50_ on the chloroquine-sensitive strain 3D7 as reported by Douanla et al. [9]. RI values below 1 indicated inhibitors preferentially acting against the resistant strain and vice versa.

#### 4.4.4. Antileishmanial Activity of Extract, Fractions, and Compounds

The antileishmanial activity of crude extracts, derived fractions, and isolated compounds against cultured *L. donovani* promastigotes was evaluated using the resazurin fluorimetric assay as described by Siqueira-Neto et al. (2010) [46]. Briefly, promastigotes from a logarithmic phase culture (4 × 10^5^ cells/mL; 90 μL) were seeded in 96-well microplates and treated with 10 μL of inhibitors at different triplicate concentrations (100 µg/mL–0.16 µg/mL for extract/fractions and 50 µg/mL–0.08 µg/mL for compounds). Plates were incubated for 28 h at 28 °C, followed by the addition of 1 mg/mL resazurin (0.15 mg/mL in DPBS; Sigma, Darmstadt, Germany). The negative and positive controls were, respectively, 0.1% DMSO and amphotericin B (Sigma, Darmstadt, Germany) (10–0.016 μg/mL) treated wells. After an additional incubation for 44 h, plates were then read on a Magelan Infinite M200 fluorescence plate reader (Tecan, Männedorf, Switzerland) at excitation and emission wavelengths of 530 and 590 nm, respectively. For each sample, growth inhibition percentages were calculated with Microsoft Excel Software; then, from the nonlinear regression fit, concentration–response curves were generated to deduce the median inhibitory concentration (IC_50_) using the GraphPad Prism 8.0 software.

#### 4.4.5. In Vitro Cytotoxicity Assay

The murine macrophages, Raw 264.7 cells were grown in a T-25 cm^2^ flask containing the complete Dulbecco’s modified Eagle’s medium (Sigma–Aldrich, Munich, Germany) supplemented with 10% FBS (Sigma–Aldrich, Munich, Germany), 1% non-essential amino acids, and 1% (*v/v*) penicillin–streptomycin (Sigma–Aldrich, Munich, Germany) and incubated at 37° in a humidified atmosphere containing 5% CO_2_. For the experiments, 80–90% confluent cell culture was trypsinized, counted using a Neubauer haemacytometer, and seeded into a 96-well cell culture treated flat-bottomed plate (SARSTEDT, Inc. Newton, NC 28658, USA) at a cell density of 10^4^ cells/well. Plates were incubated overnight to allow cell adhesion. Cells were then treated with serially diluted concentrations of crude extract, fractions, and isolated compounds for 48 h. Podophyllotoxin (Sigma–Aldrich, Munich, Germany) was used as positive control with the highest concentration of 10 μM. Afterwards, 10 μL of resazurin solution (0.15 mg/mL prepared in PBS) was added to each well, and the plates were additionally incubated for 4 h. Of important note, resazurin is a nonfluorescent, nontoxic, and cell-permeable blue dye which is used to quantify cell viability following its conversion into the highly fluorescent pink dye resorufin in response to changes in the reducing environment within the cytosol of the cell. The level of fluorescence that positively correlates with cell viability was measured using a Magellan Infinite M200 plate reader (Tecan, Germany) at an excitation wavelength of 530 nm and an emission wavelength of 590 nm. The growth proliferation percentage of cells was calculated with respect to the negative control. The concentration–response curves were plotted using inhibitory percentages versus the logarithm of the drug concentration to determine the concentration of drug that reduces cell viability by 50% (CC_50_) using GraphPad Prism 8.0 software. Selectivity indices (SI) [SI = CC_50_/IC_50_] were determined for each inhibitor based on their antiparasitic activity (IC_50_) and cell cytotoxicity (CC_50_ on Raw cells). Samples with SI greater than 10 were classified as poorly toxic toward Raw cells [47].

## 5. Conclusions

The chemical study of the stem bark of *Antrocaryon klaineanum* led to the isolation and characterized of six (**1**–**6**) secondary metabolites, among which an unreported glucocerebroside derivative antroklaicerebroside (**1**). The chemical result is in agreement with the chemotaxonomy of *Antrocaryon klaineanum* belonging to the *Antrocaryon* genus and to the Anacardiaceae family, while the biological results are in accordance with the previous literature as well as the traditional uses of this plant species in the treatment of hemorrhoid and malaria.

## Figures and Tables

**Figure 1 molecules-28-02730-f001:**
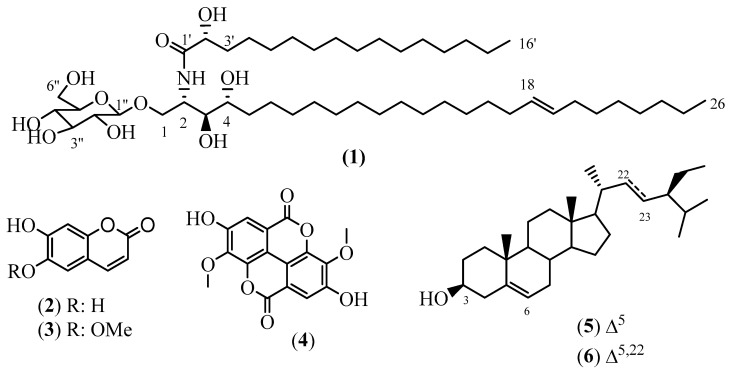
Structures of compounds **1**–**6**.

**Figure 2 molecules-28-02730-f002:**
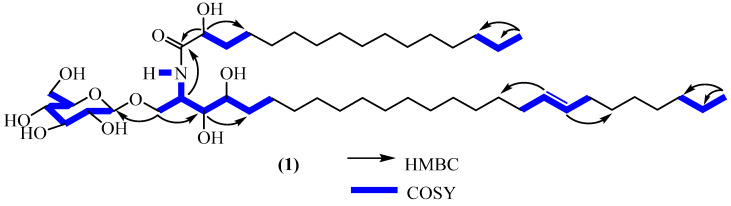
Important COSY and HMBC correlations of compound **1**.

**Figure 3 molecules-28-02730-f003:**
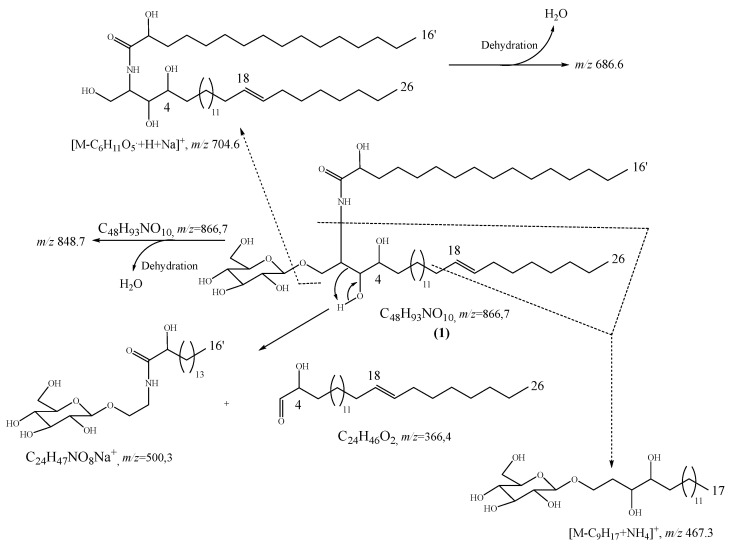
Main fragmentations based on the MS/MS spectrum of compound **1**.

**Figure 4 molecules-28-02730-f004:**
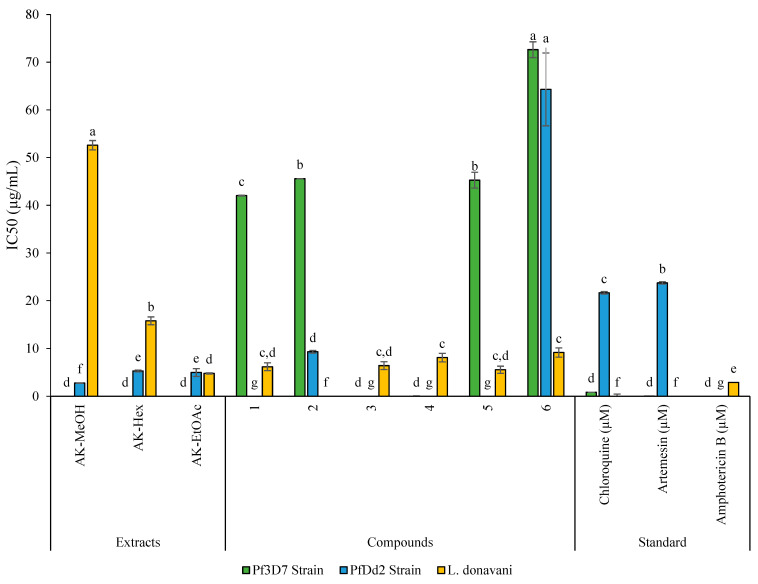
Antiplasmodial activity against sensitive chloroquin 3D7 and Dd2 strains resistant to chloroquine and antileishmanial activity of extracts and compounds. The results with the same letter are not significantly different at *p* ≤ 0.05 (Duncan’s test).

**Figure 5 molecules-28-02730-f005:**
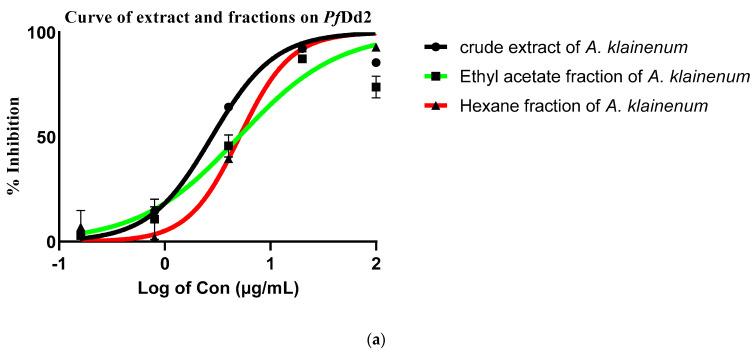
(**a**) Curve of extract and fractions of antiplasmodial activity against chloroquine-resistant Dd2; (**b**) Dose-response curve of control of antiplasmodial activity on chloroquine-resistant Dd2.

**Table 1 molecules-28-02730-t001:** ^13^C, ^1^H, and HMBC NMR spectroscopic data (^1^H 600 MHz, ^13^C 150 MHz) for compounds **1.** (In Acetone-*d*_6_ and CD_3_OD).

N°	Antroklaicerebroside (1)				
	Acetone-*d*_6_		CD_3_OD		Acetone-*d*_6_ and CD_3_OD
	*δ* _C_	*δ*_H_ (m, *J* in Hz)	*δ* _C_	*δ*_H_ (m, *J* in Hz)	^2,3^*J*_HC,_ HMBC
**1**	70.2	3.96 (dd, 10.9, 6.5) 3.90 (dd, 10.9, 4.5)	69.9	4.05 (dd, 10.7, 6.2) 3.82 (dd, 10.7, 3.9)	2, 3, 1″
**2**	51.7	4.30 (m)	51.6	4.27 (m)	3, 1′
**3**	75.9	3.56 (m)	75.6	3.60 (t, 6.0)	1, 2, 4, 5
**4**	72.5	3.58 (m)	72.9	3.53 (m)	2
**5**	33.6	1.42 (m)	32.9	1.42 (m)	
**6**	28.0	2.06 (m)	28.3	2.06 (m)	
**7–16**	30.5	1.29 (m)	30.8	1.29 (m)	
**17**	33.4	1.98 (m)	33.1	1.96 (m)	18, 19
**18**	131.2	5.41 (m)	131.4	5.42 (brt, 5.2)	
**19**	130.8	5.35 (m)	130.9	5.37 (m)	
**20**	33.5	2.01 (m)	33.8	1.98 (m)	18, 19
**21–23**	30.5	1.29 (m)	30.8	1.29 (m)	
**24**	32.6	1.29 (m)	32.6	1.29 (m)	
**25**	23.5	1.30 (m)	23.7	1.32 (m)	
**26**	14.5	0.88 (t, 7.0)	14.5	0.88 (t, 7.0)	24, 25
**1′**	nd	-	177.2	-	
**2′**	72.6	4.04 (m)	73.0	4.02 (dd, 7.7, 3.8)	1′, 3′, 4′
**3′**	35.6	1.77 (m) 1.60 (m)	35.7	1.77 (m) 1.60 (m)	1′, 2′
**4′**	25.9	1.44 (m)	25.9	1.44 (m)	
**5′–13′**	30.5	1.29 (m)	30.8	1.29 (m)	
**14′**	32.6	1.29 (m)	32.6	1.29 (m)	
**15′**	23.5	1.30 (m)	23.5	1.32 (m)	
**16′**	14.5	0.88 (t, 7.0)	14.5	0.88 (m)	14′, 15′
**1″**	104.9	4.34 (d, 7.7)	104.7	4.29 (d, 7.8)	1, 2″
**2″**	74.9	3.16 (brt, 8.0)	75.0	3.18 (dd, 9.1, 7.8)	1″, 3″
**3″**	78.0	3.38 (m)	78.0	3.38 (t, 9.1)	4″
**4″**	71.6	3.32 (m)	71.6	3.28 (m)	3″
**5″**	77.8	3.32 (m)	77.9	3.28 (m)	
**6″**	63.0	3.85 (dd, 11.5, 6.5) 3.65 (dd, 11.5, 5.9)	62.7	3.87 (dd, 11.8, 1.7) 3.67 (dd, 11.8, 5.4)	4″, 5″
**-NH**	-	7.51 (d, 8.9)	-	nd	

δ (ppm), nd: not determinated.

**Table 2 molecules-28-02730-t002:** Antiplasmodial, antileishmanial, and cytotoxicity parameters of crude extract, fractions, and compounds from *A. klaineanum*.

Samples	Cytotoxicity Raw Cells	*P. falciparum*					*L. donovani* Promastigote	
	CC_50_ ± SD (µg/mL)	*Pf*3D7 IC_50_ ± SD (µg/mL)	SI_*Pf*3D7	*Pf*Dd2 IC_50_ ± SD (µg/mL)	SI_*Pf*Dd2	RI	IC_50_ ± SD (µg/mL)	SI
Extract								
**AK-MeOH**	>500	>100	ND	2.78 ± 0.06	>179.86	-	>100	ND
**Fractions**								
**AK-Hex**	>500	>100	ND	5.30 ± 0.19	>94.34	-	15.78 ± 0.82	>31.69
**AK-DCM**	>500	>100	ND	>100	ND	ND	>100	ND
**AK-EtOAc**	>500	>100	ND	4.98 ± 0.81	>100.40	-	4.80 ± 0.13	>104.17
Compounds								
Antroklaicerebroside **(1)**	>500	42.04 ± 0.08	>11.72	>50	ND	>1.19	6.19 ± 0.79	>80.78
Esculetin **(2)**	>500	45.57 ± 0.08	>10.97	9.30 ± 0.29	>53.76	0.2	>50	ND
Scopoletin **(3)**	>500	>50	ND	>50	ND	ND	6.40 ± 0.80	>78.13
3,3′-Dimethylellagic acid **(4)**	>500	>50	ND	>50	ND	ND	8.07 ± 0.90	>61.96
*β*-sitosterol **(5)**	>500	45.27 ± 1.66	>11.04	>50	ND	>1.1	5.56 ± 0.74	>89.93
Stigmasterol **(6)**	>500	>50	ND	>50	ND		9.14 ± 0.96	>54.70
References drugs								
**Chloroquine (µM)**	-	0.85± 0.01	-	21.65 ± 0.25	-	-	-	-
**Artemesin (µM)**	-	0.02 ± 0.00	-	23.73 ± 0.24	-	-	-	-
**Amphotericin B (µg/mL)**	-	-	-	-	-	-	2.90 ± 0.46	-
**Podophyllotoxin (µM)**	0.71 ± 0.20	-	-	-	-	-	-	-

**AK-MeOH**: methanol extract of *Antrocaryon klaineanum*, **AK-Hex**: *n*-hexane fraction of *Antrocaryon klaineanum* extract, **AK-DCM**: dichloromethane fraction of *Antrocaryon klaineanum* extract, **AK-EtOAc**: Ethylacetate fraction of *Antrocaryon klaineanum* extract, SI: Selectivity index, ND: not determinated, NC: not calculated, *P. falciparum.*

## Data Availability

Data are contained within the article and Appendix A.

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
