# Peer review of "Antileishmanial and Antiplasmodial Activities of Secondary Metabolites from the Root of Antrocaryon klaineanum Pierre (Anacardiaceae)"

_molecules, 2023, doi:10.3390/molecules28062730_

Round 1
Reviewer 1 Report
The manuscript entitled ¨ Antileishmanial and antiplasmodial activities of secondary metabolites
from the root of Antrocaryon klaineanum Pierre (Anacardiaceae)¨, by Ngnoung et al. addresses aspects related to the identification of molecules with antiparasitic potential, specifically against Leishmania donovani and Plasmodium falciparum. The Manuscript presents interesting results, even so, the results are presented in a somewhat confusing way and must be considerably improved to be admitted for publication.
The summary must be reformulated, since it already begins reporting results. The objective and conclusion are not clear in the abstract.
In the introduction: Line 56-63: Very long text and only one reference used.
In order to improve the justification of the work carried out, it is suggested to address in a slightly more detailed way epidemiological aspects in Cameroon of the diseases under study, as well as the difficulties encountered in the treatment of diseases.
It is not clear if the leishmanicidal activity was determined by colorimetry or it was a fluorimetric assay (see methodology).
Perhaps the results can be merged with the discussion, so that the reading is more fluid. From the way presented it is not clear what is the discussion and what is the presentation of the results.
Organize list of references according to the parameters of the journal
Author Response
Please see point by point response file!

Reviewer 2 Report
Type of the Paper (Article)
Title: Antileishmanial and antiplasmodial activities of secondary metabolites from the root of Antrocaryon klaineanum Pierre (Anacardiaceae)
The overall article was well written except for some typo errors.
Minor error:
1) Comments: Page 3, Line 101: ([?]58620 = +0.2 (c 2.5, Acetone)), instead of: ([?]58620 = +0.2 (c 2.5,Acetone)); a space is need between 2.5 and Acetone
2) Comments: Page 3, Line 103: (calcd. m/z 866.6692 for C48H93NO10Na) instead of: (calcd. m/z 866.6692 for C48H93NO10Na+).
3) Comments: Page 4, Line 140 and Figure 3: [M+Na]+ m/z 500.3 for C24H47NO3Na is not match the molecular formula C24H47NO3Na also the figure 3. (It seems some protons are missing while counting for C24H42NO8Na )
4) Suggestion ROESY analysis need: Absence of correlation from H-2 and H-3 in ROESY spectrum will more support the 2S and 3S configuration.

Author Response

(The authors gave the same response as above.)

Round 2
Reviewer 1 Report
The authors of the manuscript answered the questions asked. In this sense, it is suggested that the work is suitable for publication.